# Highly Efficient NO_2_ Sensors Based on Al-ZnOHF under UV Assistance

**DOI:** 10.3390/ma16093577

**Published:** 2023-05-07

**Authors:** Xingyu Yao, Rutao Wang, Lili Wu, Haixiang Song, Jinbo Zhao, Fei Liu, Kaili Fu, Zhou Wang, Fenglong Wang, Jiurong Liu

**Affiliations:** 1Key Laboratory for Liquid-Solid Structural Evolution and Processing of Materials, Ministry of Education and School of Materials Science and Engineering, Shandong University, Jinan 250061, China; xinghuiyxy@gmail.com (X.Y.);; 2Henan Joint International Research Laboratory of Nanocomposite Sensing Materials, School of Chemical and Environmental Engineering, Anyang Institute of Technology, Anyang 455000, China; 3School of Materials Science and Engineering, Qilu University of Technology (Shandong Academy of Sciences), Jinan 250353, China

**Keywords:** gas sensors, Al^3+^-doped, ZnOHF, NO_2_, UV assistance

## Abstract

Zinc hydroxyfluoride (ZnOHF) is a newly found resistive semiconductor used as a gas-sensing material with excellent selectivity to NO_2_ because of its unique energy band structure. In this paper, Al^3+^ doping and UV radiation were used to further improve the gas-sensing performance of ZnOHF. The optimized 0.5 at.% Al-ZnOHF sample exhibits improved sensitivity to 10 ppm NO_2_ at a lower temperature (100 °C) under UV assistance, as well as a short response/recovery time (35 s/96 s). The gas-sensing mechanism demonstrates that Al^3+^ doping increases electron concentration and promotes electron transfer of the nanorods by reducing the bandgap of ZnOHF, and the photogenerated electrons and holes with high activity under UV irradiation provide new reaction routes in the gas adsorption and desorption process, effectively promoting the gas-sensing process. The synergistic effect of Al^3+^ and UV radiation contribute to the enhanced performance of Al-ZnOHF.

## 1. Introduction

As a typical pollution gas, nitrogen dioxide (NO_2_) seriously threatens human health and the environment [1,2]. Exposure to NO_2_ with a concentration of over 200 μg/m^3^ in the short term or over 40 μg/m^3^ in the long term will pose a serious threat to the normal operation of the respiratory system [3]. Additionally, NO_2_ is also the main source of acid rain and photochemical smog. Hence, the development of gas sensors for efficiently detecting the concentration of NO_2_ in the atmosphere is highly significant and essential. In the research on NO_2_ sensors, resistance gas sensors stand out among different kinds of gas sensors due to their excellent sensitivity, selectivity, and stability [4,5,6], as well as their relatively low cost and easy operation [7]. As the core part of sensors, the sensing materials directly determine the performance of gas sensors.

Zinc hydroxyfluoride (ZnOHF) is a new type of semiconductor discovered in recent years which has gradually attracted the attention of researchers in various fields, such as photocatalysts [8,9], ultrasonic degradation [10] and solar cells [11], for its unique crystal and energy band structure [12,13]. In our previous research, we found that ZnOHF is an excellent gas-sensing material with good selectivity to NO_2_ gas [14]. The unique energy band structure of ZnOHF inhibits the adsorption and dissociation of oxygen molecules on its surface, and the little absorbed oxygen species on the surface of ZnOHF lead to its wide detection range to NO_2_, which makes ZnOHF a good candidate for NO_2_ detection among resistive semiconductors. However, the gas sensors based on ZnOHF still have room for improvement in performance; for example, the operating temperature of ZnOHF-based gas sensors is still high, and its sensitivity still has a certain gap compared with many other NO_2_-sensing materials. Therefore, further improvement for ZnOHF to obtain better sensing performance is necessary.

Light-activated gas sensors have been widely studied as an effective improvement method in recent years. Under the assistance of light irradiation, there will be more electrons and holes with high activity generated on the surface of materials, which is beneficial to the adsorption and desorption of target gases [15,16]. Wang et al. [17] found that ZnO nanoplates exhibited a great enhancement in sensitivity and recovery time in NO_2_ detection under intermittent UV irradiation. This phenomenon can be explained by the competitive adsorption of NO_2_ and O_2_ and the modulating action of gas adsorption. Zhang and his group [18] grew TiO_2_ nanoplates in situ with a highly active crystal plane on MXene (Ti_3_C_2_T_x_) and obtained greatly enhanced gas-sensing performance to NH_3_ under UV light. The density functional theory revealed that this composite structure exhibits the highest adsorption affinity to NH_3_ with UV light assistance. Additionally, Tai’s group [19] also applied gamma-ray in NO_2_ detection, and they found that it can effectively enhance the sensitivity of ZnO nanorods at room temperature by constructing lots of defects. Bang et al. [20] presented a proton-beam irradiation method to engineer surface-point defects on ZnO in order to improve the preferential adsorption on the surface compared to water molecules so that it can detect NO_2_ with less humidity interference. As a kind of photocatalyst, ZnOHF has excellent photochemical properties, including suitable band gaps and strong light absorbance, which makes it possible to act as a better gas sensor of NO_2_ under UV radiation [21,22].

Ion doping is a promising approach in gas-sensing performance improvement. On the one hand, doping ions with different quantities of charges can effectively adjust the carrier concentration and energy band structure of semiconductors [23,24]. On the other hand, due to the difference in diameters between doping ions and original ions, ion doping can create lots of lattice imperfections. Some imperfections can act as efficient adsorption sites of oxygen molecules or target gas molecules, thus improving their gas-sensing performance [25]. Qin and his colleague [26] synthesized metal–organic-frameworks-derived Mn-doped Co_3_O_4_ porous nanosheets, which exhibited better CO-sensing performance than pure Co_3_O_4_ because of the oxygen vacancies generated in the ion doping process. Mokrushin et al. [27] doped Eu and Pr ions into ZnO powders and found that it not only improved the sensitivity but also obtained great humidity resistance performance. Cr^3+^-doped In_2_O_3_ was prepared by Sun et al. [28], and this material shows high sensitivity, a low detection limit, and short response/recovery time in NH_3_ sensing due to the increase in active sites and reduced material resistance. Al^3+^ ion is one of the common doping ions utilized in zinc-based materials’ modification. Since Al^3+^ has a higher charge than Zn^2+^, Al^3+^ doping can increase electron concentration, which will benefit the gas-sensing process [29,30]. Moreover, Al^3+^ has a similar diameter to Zn^2+^, indicating that Al^3+^ doping will not seriously damage the crystal structure of the host ZnOHF [31,32].

Herein, Al-ZnOHF nanorods with optimized Al^3+^ concentrations were prepared through a simple hydrothermal method. The NO_2_-sensing performance of Al^3+^-doped samples is significantly improved compared with pure ZnOHF under UV irradiation with a wavelength of 395 nm and a strength of 6 mW/cm^2^. Additionally, especially, the 0.5 at.% Al-ZnOHF sample exhibits the best gas-sensing properties. The NO_2_-sensing mechanism analysis revealed that the bandgap narrowing and improvement in UV absorbance after Al^3+^ doping are the main reasons for its performance enhancement.

## 2. Materials and Methods

### 2.1. Chemicals

All chemicals utilized in this work, including zinc acetate (Zn(CH_3_COO)_2_•2H_2_O), ammonium fluoride (NH_4_F), aluminum nitrate (Al(NO_3_)_3_•9H_2_O), and hexamethylenetetramine (HMT), were purchased from Sinopharm Chemical Reagent Co., Ltd. (Shanghai, China) and were of analytical grade, which can be used without any purification.

### 2.2. Synthesis of Al^3+^-doped ZnOHF

Al-ZnOHF nanorods with different molar ratios of Al^3+^ (0 at.%, 0.1 at.%, 0.2 at.%, 0.5 at.%, and 1 at.%) were synthesized via a simple hydrothermal method. Initially, 0.65 g Zn(CH_3_COO)_2_•2H_2_O, 0.42 g HMT and 0.051 g NH_4_F were dissolved in 15 mL of distilled water successively with continuous stirring to form a homogeneous solution. Different amounts of Al(NO_3_)_3_•9H_2_O were then added to the mixture with continuous stirring. After stirring for another 0.5 h, the solution was transferred into 20 mL Teflon-lined autoclaves and heated at 368 K for 2 h. The products were collected, purged with distilled water and ethyl alcohol, and then dried at 340 K for 6 h.

### 2.3. Characterizations

The crystal structures of samples were tested through X-ray powder defection (XRD, DMAX-2500PC, Regaku, Japan) equipped with Cu-Kα (λ = 1.542 Å) radiation with a step size of 0.02° and a scan rate of 10°/min in the range of 10–90°. The surface morphologies and micro-structure characterizations were carried out by scanning electron microscopy (SEM, SU-70, Hitachi, Japan) operated at an accelerating voltage of 15 kV and transmission electron microscopy (TEM, JEM 2100, JEOL Ltd., Akishima, Japan) operated at 200 kV. X-ray photoelectron spectrometer (XPS, AXIS Supra, Kratos, Japan) measurement (equipped with monochromatic Al-Kα radiation, hυ = 1486.6 eV) was utilized to analyze the valence state of elements and the chemical environment of atoms on the surface of samples. The UV-vis diffuse reflection spectrum (UV-vis DRS) was recorded on Agilent Cary 300/PE lambda 750S (America) ultraviolet spectrophotometer and was used to test UV absorption and bandgap of the samples.

### 2.4. Gas-Sensing Properties Test

The preparation method of gas-sensing units was explained as follows. First, the as-prepared samples were dispersed into distilled water with a weight ratio of 1:5 and treated with ultrasound for 10 min. Additionally, the slurry was dropped on the Al_2_O_3_ substrate, which has four Pt wires as the electrodes, and was dried at 80 °C. After repeating this step 2–4 times, the substrate was annealed at 200 °C for 3 h and then the electrodes were welded on the pedestal to form a gas-sensing unit. After that, the sensor was aged at 4 V for 7 days before the gas-sensing test. The gas-sensing properties of gas sensors based on various samples were tested via a WS-30B gas sensitivity instrument (Zhengzhou Winsen Electronics Co., Ltd., Zhengzhou, China) at an environmental temperature of 25 °C and humidity of 20–30%. The UV light was provided by LED light with wavelength of 395 nm (Ceaulight Co., Beijing, China). The response value of sensors (S) is calculated as S = R_g_/R_a_ for oxidizing gases and S = R_a_/R_g_ for reducing gases, where R_a_ and R_g_ are the resistance of the sensor in air and target gases, respectively. Additionally, the response and recovery times are defined as the times from gas input or pumped out until the response value changes to 90% of its maximum value.

## 3. Results and Discussion

### 3.1. Characterization

Figure 1 shows the XRD patterns of Al-ZnOHF with different amounts of aluminum ions. The diffraction peaks of each sample could be approximately matched with the standard data of orthorhombic-phase ZnOHF (JCPSD No: 74-1816). After the addition of Al^3+^, the central position of diffraction peaks changed significantly. It can be seen in the magnified region around the strongest peak (Figure 1b) that the peak trended to a higher angle with the increase in Al^3+^ concentration at first, indicating that the Al^3+^ with the size of 0.0535 nm may occupy some positions of the Zn^2+^ with the size of 0.074 nm. Additionally, then, the peak turned back to the lower angle (1 at.% Al-ZnOHF), indicating that the Al^3+^ dopes into the interstitial void of the ZnOHF lattice, which results in the broadening of the interplanar crystal spacing.

Figure 2a,b shows the SEM images of pure ZnOHF. The basic structure of pure ZnOHF is nanorods, and the ends of these nanorods are gathered together to constitute flower-like structures with sizes ranging from 3 to 5 μm. SEM graphs of samples with different Al^3+^ dopings are exhibited in Figure 2c–f. After Al^3+^ doping, the flower-like structures of pure ZnOHF change into irregular bundles, which also consist of nanorods. With the increase in the amount of Al^3+^, the length of bundles gradually shortened and thickened, and the morphology gradually changed to microbelts. Additionally, lots of small nanoplates also appeared, and the quantity gradually increased with the increase in Al^3+^ content. Finally, when the Al^3+^ doping amount reached 1 at.%, the morphology of the sample was mainly composed of long microbelts and plenty of nanoplates.

To further analyze the microstructure of the obtained samples, TEM and HRTEM images of pure ZnOHF and 0.5 at.% Al-ZnOHF are shown in Figure 3. As shown in Figure 3a,b, pure ZnOHF exhibits a regular nanorods-assembled flower-like structure. The HRTEM image (Figure 3c) shows that the diameter of nanorods is about 14 nm. Additionally, the lattice fringe spacing of pure ZnOHF could be measured as 0.25 nm, corresponding to the (1 1 1) crystal plane of ZnOHF. After Al^3+^ doping, the morphology of samples changed greatly (Figure 3d,e), which is in accord with the SEM images. Instead of forming a flower-like structure, the nanorods were clustered into bundles and scattered irregularly. The HRTEM image of 0.5 at.% Al-ZnOHF (Figure 3f) exhibits that the diameter of nanorods slightly increased to 25 nm. Its lattice fringe spacing was calculated to be 0.42 nm, a little smaller than that of the (1 1 0) crystal plane. Hence, Al^3+^ doping changed the exposed surface crystal plane of samples and reduced the crystalline interplanar spacing.

The surface chemical properties of 0.5 at.% Al-ZnOHF were determined via X-ray photoelectron spectroscopy (XPS). As shown in Figure 4a, all 3 main elements (Zn, O, F) can be easily found in the wide XPS spectrum of both pure ZnOHF and 0.5 at.% Al-ZnOHF, and there is also a tiny peak at about 75 eV in the spectra of 0.5 at.% Al-ZnOHF, which can be matched with the Al element. The Zn 2p spectrum in Figure 4b shows that there are 2 peaks centered at 1021.8 eV and 1044.8 eV, corresponding to the Zn^2+^ orbits, 2p3/2 and 2p1/2, respectively [33]. The peaks of F 2p (Figure 4c) of these 2 samples are centered at about 684.8 eV, matching well with F^−^ in the crystal lattice [34]. By comparison, the peaks of Zn and F of these two samples are approximately the same, meaning that Al^3+^ doping does not change the chemical environment of Zn and F atoms. The difference between these two samples at the high-resolution XPS regions of Al 2p and O 1s is shown in Figure 4d–e. The 0.5 at.% Al-ZnOHF exhibits a significant peak at about 75.2 eV, which can be indexed with Al^3+^ [35], while pure ZnOHF just shows a gentle background curve in this region (Figure 4d). This difference confirmed that Al ions were successfully doped into ZnOHF. The O 1s peaks of both pure ZnOHF and 0.5 at.% Al-ZnOHF can be deconvoluted into two Gaussian peaks, which are matched with lattice oxygen and adsorbed oxygen species, respectively (Figure 4e) [14,36]. It is significant that the proportion of adsorbed oxygen species of 0.5 at.% Al-ZnOHF is higher than that of pure ZnOHF, confirming that Al^3+^ doping promotes the surface adsorption of oxygen. In addition, the peaks of the 2 oxygen species exhibit a slight blueshift after Al^3+^ doping (O_latt._ move from 532.0 eV to 531.6 eV and O_ads._ move from 532.8 eV to 532.2 eV). This phenomenon may be ascribed to the weaker adsorption capacity for electrons of Al^3+^ than that of Zn^2+^. When Al^3+^ substitutes the Zn^2+^ in the lattice, the electron density increases due to the construction of the Al-O bond [37,38,39]. Therefore, Al^3+^ doping can decrease the binding energy of electrons of oxygen species and obtain adsorbed oxygen with higher activity.

### 3.2. Gas-Sensing Properties

The gas-sensing properties of ZnOHF with various amounts of Al^3+^ doping were tested under different temperatures (Figure 5a). Due to the low thermal stability of ZnOHF at high temperatures (it decomposes into ZnO and HF at over 300 °C) [14], the test temperature should be controlled under 300 °C. Significantly, the best operating temperatures of all samples are 160 °C. The response value of Al^3+^-doped samples shows various levels of increase compared with pure ZnOHF. Especially, gas sensors based on 0.5 at.% Al-ZnOHF obtained the highest response (49.43) to 10 ppm NO_2_, 2.42 times higher than that of pure ZnOHF. In addition, the dynamic response curves of sensors based on 0.5 at.% Al-ZnOHF and pure ZnOHF to 10 ppm NO_2_ are shown in Figure 5b. It is worth noting that the response time of 0.5 at.% Al-ZnOHF (50 s) is a little shorter than that of pure ZnOHF (112 s), but both of them exhibit extremely long recovery times or even cannot recover back to the previous resistance. The resistance of samples at different temperatures is shown in Appendix A. It is evident that with the increase in the Al^3+^ doping amount, the resistance of samples rises at first, probably due to the destruction of the regular morphology of pure ZnOHF after ion doping, which is not conducive to electron conduction. Additionally, the resistance reduces when the Al^3+^ doping amount continuously increases because of the increase in carrier concentration. The extremely high resistance of samples also seriously restricts the gas-sensing test at low temperatures.

To solve this problem, UV-light radiation was utilized in the gas-sensing process. It can be seen in Appendix A that the resistance of all samples significantly decreases under UV radiation, and the Al^3+^-doped samples obtained a higher reduction rate with higher UV absorbance. The gas-sensing properties of samples under UV light with the wavelength of 395 nm and strength of 6 mW/cm^2^ at different temperatures were tested, as illustrated in Figure 5c. It is significant that the best operating temperature of 0.1, 0.2, and 0.5 at.% Al-ZnOHF is 100 °C, while that of 1 at.% Al-ZnOHF is 80 °C, indicating that the operating temperature decreased after UV radiation. Specifically, the sensor based on 0.5 at.% Al-ZnOHF exhibited the highest response, with a value of 110.83 to 10 ppm NO_2_, which is approximately 4 times higher than that of pure ZnOHF (25.29). Additionally, the best response values of 0.1 at.%, 0.2 at.%, and 1 at.% Al-ZnOHF are 51.32, 64.55, and 35.22, respectively. Therefore, as the concentration of Al^3+^ goes up, the best operating temperature of samples exhibits a downtrend, and the sensitivity of ZnOHF increases initially and then decreases. Compared with the test results without UV light, the response of 0.5 at.% Al-ZnOHF to 10 ppm NO_2_ under UV light exhibit over 2-fold promotion, while the responses of other samples only increased a little. As shown in Figure 5d, the response/recovery times of pure ZnOHF and 0.5 at.% Al-ZnOHF were 83 s/128 s and 35 s/96 s, respectively. Both of them were significantly reduced compared with that in the dark environment. Therefore, the 0.5 at.% Al-ZnOHF sample obtains a high response and fast response/recovery speed at 100 °C with UV assistance.

The dynamic sensing curve of 0.5 at.% Al-ZnOHF to NO_2_ with various concentrations is shown in Figure 6a. With the increase in the gas concentration, the response value continuously rises, and the response/recovery time exhibits no apparent change. It is worth noting that the detection limit of 0.5 at.% Al-ZnOHF is 0.25 ppm, with a response value of 2.18, which is higher than that of pure ZnOHF, as shown in Appendix A (1.57 to 0.25 ppm NO_2_). Additionally, the dynamic response curve of these two samples in the dark (Appendix A) exhibits that the detection limits of pure ZnOHF and 0.5 at.% Al-ZnOHF in dark are 0.5 ppm. Therefore, UV light effectively reduces the detection limit of Al-ZnOHF samples. The relationship between NO_2_ concentration and the response value of pure ZnOHF and 0.5 at.% Al-ZnOHF is shown in Figure 6b, respectively. Significantly, the logarithm of gas concentration and response value of both samples shows an excellent linear relationship with high reliability. Additionally, the slope of the fitted curve of 0.5 at.% Al-ZnOHF is higher than that of pure ZnOHF. This phenomenon means that when exposing a sensor based on 0.5 at.% Al-ZnOHF to NO_2_; a slight change in gas concentration will lead to a more dramatic change of response value than that of pure ZnOHF, which can be explained by the following equation.
$$\frac{d\left[S\right]}{d\left[C\right]}=\frac{S}{C}\;\frac{dln\left[S\right]}{dln\left[C\right]}$$

Therefore, the gas sensor based on 0.5 at.% Al-ZnOHF can detect NO_2_ more sensitively and accurately than pure ZnOHF.

Repeatability is also an essential feature of gas sensors. The dynamic sensing curve of 0.5 at.% Al-ZnOHF in 4 cycles was tested. As shown in Figure 6c, the response value and response/recovery time show no discernible difference in these cycles. Therefore, the 0.5 at.% Al-ZnOHF sensor exhibits superior repeatability to NO_2_ gas at 100 °C under UV assistance. Moreover, the response values of 0.5 at.% Al-ZnOHF within one month remained relatively stable, as shown in Figure 6d, proving its great stability. In order to understand the interference of other gases in practical application, the gas-sensing test of the samples to other gases is shown in Figure 6e. Sensors based on both pure ZnOHF and 0.5 at.% Al-ZnOHF exhibit a high response to NO_2_, while the response values to other gases are lower than 2, indicating that both samples show excellent selectivity. Compared with many other zinc-based gas-sensing materials shown in Table 1, the Al-ZnOHF sample exhibits a great gas-sensing performance (high sensitivity, short response/recovery time, and low detection limit) at relatively low temperatures.

In practical application, the relative humidity of the gas testing environment is an important factor that affects gas-sensing properties at low operating temperatures. Relative humidity is defined as the ratio of the absolute humidity in the air to the saturated vapor pressure of water under the same condition. Figure 7a shows the sensing curves of the 0.5 at.% Al-ZnOHF sample at various relative humidities at 100 °C. Before the relative humidity reaches 50%, the response value of 0.5 at.% Al-ZnOHF remains relatively stable. When the relative humidity continues to rise, the response value shows a somewhat decline (Figure 7b). However, the response value of the sensor at RH95 was 73.79, still keeping about 67.0% of that at RH20. Figure 7c illustrates that the higher the relative humidity of the test environment is, the faster the response value’s decline. In terms of the response time, with the relative humidity increasing, it rises a little while the recovery time increases to about 250 s (Figure 7c). Thinking of the low operating temperature of 100 °C, Al-ZnOHF exhibits a relatively great water-resistance performance, especially under relative humidity lower than RH50.

### 3.3. Gas-Sensing Mechanism

As an n-type semiconductor, the resistance of ZnOHF is mainly controlled by electrons in the conductive band. According to reference research, the resistance of gas-sensing material is mainly determined by the electron depletion layer on the surface, including its resistance and width [45,46]. Under relatively low temperatures, when exposing sensing material to air without photoactivation, some oxygen molecules adsorb on the surface and consume electrons to form oxygen ions ($O_{2\left(ads\right)}{}^{-}$, 0–150 °C), as shown in Reaction (1), thereby forming an electron depletion layer with high resistance [47]. When UV light is applied, for one thing, the surface adsorbed oxygen ions react with the photogeneration holes to release oxygen (Reaction (2)). For another, the physically adsorbed oxygen molecules capture photogeneration electrons to form new oxygen species with high activity (Reaction (3)). When these two reactions reach a balance, the surface of the material is coated with photoactivated oxygen ions [48].
(1)$$O_{2\left(gas\right)}+e^{-}\to O_{2\left(ads\right)}{}^{-}$$
(2)$$O_{2\left(ads\right)}{}^{-}+h_{h\nu }{}^{+}\to O_{2\left(gas\right)}$$
(3)$$O_{2\left(gas\right)}+e_{h\nu }{}^{-}\to O_{2\left(h\nu \right)}{}^{-}$$

The NO_2_-sensing process is shown in Figure 8. When exposing Al-ZnOHF sensors to NO_2_ without UV light, the NO_2_ molecules are ionized into ${\mathrm{NO}}_{2}{}^{-}$ by capturing conductive band electrons, leading to an increase in resistance (Reaction (4)) [49]. In comparison, the UV-activated electrons are more active than those in the dark, making them easier to combine with NO_2_ molecules (Reaction (5)). In addition, another reaction route will occur under UV light, as shown in Reaction (6). Some NO_2_ molecules also react with photoactivated oxygen ions and conductive band electrons to create ${\mathrm{NO}}_{3}{}^{-}$ [50]. Therefore, more NO_2_ will be adsorbed and reacted on the surface of Al-ZnOHF, and the sensitivity of the sensors demonstrates enormous improvement.
(4)$$NO_{2\left(gas\right)}+e^{-}\to NO_{2\left(ads\right)}{}^{-}$$
(5)$$NO_{2\left(gas\right)}+e_{h\nu }{}^{-}\to NO_{2\left(h\nu \right)}{}^{-}$$
(6)$$2NO_{2\left(gas\right)}+O_{2\left(h\nu \right)}{}^{-}+e_{h\nu }{}^{-}\to 2NO_{3\left(h\nu \right)}{}^{-}$$

When NO_2_ is pumped out, the surface-adsorbed ions are desorbed and release electrons to the conductive band of materials. The difference is that materials under UV light have a lot of photogenerated holes with high activity, which can efficiently promote the oxidation of ${\mathrm{NO}}_{2}{}^{-}$ and ${\mathrm{NO}}_{3}{}^{-}$ (Reactions (7) and (8)), as well as their desorption. Hence, UV light assistance can effectively accelerate the recovery process and reduce the recovery time [51,52].
(7)$$NO_{2\left(h\nu \right)}{}^{-}+h_{h\nu }{}^{+}\to NO_{2\left(gas\right)}$$
(8)$$2NO_{3\left(h\nu \right)}{}^{-}+2h_{h\nu }{}^{+}\to 2NO_{2\left(gas\right)}+O_{2\left(gas\right)}$$

In order to figure out the bandgap and UV light absorption of different samples, the UV-visible diffuse reflectance spectra of Al-ZnOHF with various Al^3+^ concentrations were tested (Figure 9a). Significantly, Al-ZnOHF exhibits a higher absorbance than pure ZnOHF, especially 0.5 at.% Al-ZnOHF. Using the T-plots of (αhν)^2^ vs. hν of samples (Figure 9b), the band gaps can be calculated as 3.31 eV for pure ZnOHF, 3.29 eV for 0.1 at.% Al- ZnOHF, 3.25 eV for 0.2 at.% Al- ZnOHF, 3.20 eV for 0.5 at.% Al-ZnOHF, and 3.40 eV for 1 at.% Al-ZnOHF. Hence, the introduction of a low content of Al^3+^ can significantly narrow the bandgap of ZnOHF, and the 0.5 at.% Al-ZnOHF sample has the minimum bandgap. The UV absorption and bandgap are the main influencing factors of the generation of photoactivated electrons and holes, thereby affecting the gas-sensing process [37,53]. Therefore, compared with other samples, 0.5 at.% Al-ZnOHF exhibits the highest increase in the response value rate before and after UV light radiation.

## 4. Conclusions

In this work, Al-ZnOHF nanorods were synthesized via a simple one-step hydrothermal method for highly efficient NO_2_-sensing detection. Under UV light assistance, the gas sensitivity of 0.5 at.% Al-ZnOHF was further enhanced, and the response/recovery time was also significantly shortened. The sample preserved good anti-humidity performance, especially under RH50. The improvement in the gas-sensing performance of Al-ZnOHF can be ascribed to the reduced bandgap and increased electron concentration. UV irradiation generated electrons and holes with high activity, which means that they can react with target gas effectively and efficiently. The new reaction route during the adsorption and desorption process also promotes the enhancement in g as-sensing performance. Therefore, this work prepares an effective gas sensor based on 0.5 at.% Al-ZnOHF for NO_2_ detection and provides new insights into modifying ZnOHF-based sensing material with superior gas-sensing performance.

## Figures and Tables

**Figure 1 materials-16-03577-f001:**
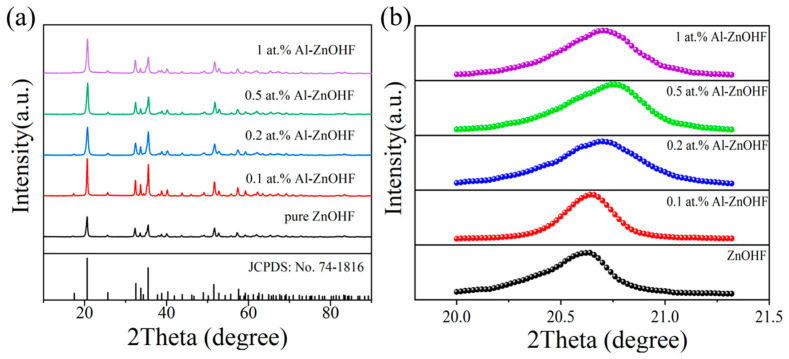
(**a**) The XRD spectra and (**b**) magnified region of the strongest peak of all Al-ZnOHF samples.

**Figure 2 materials-16-03577-f002:**
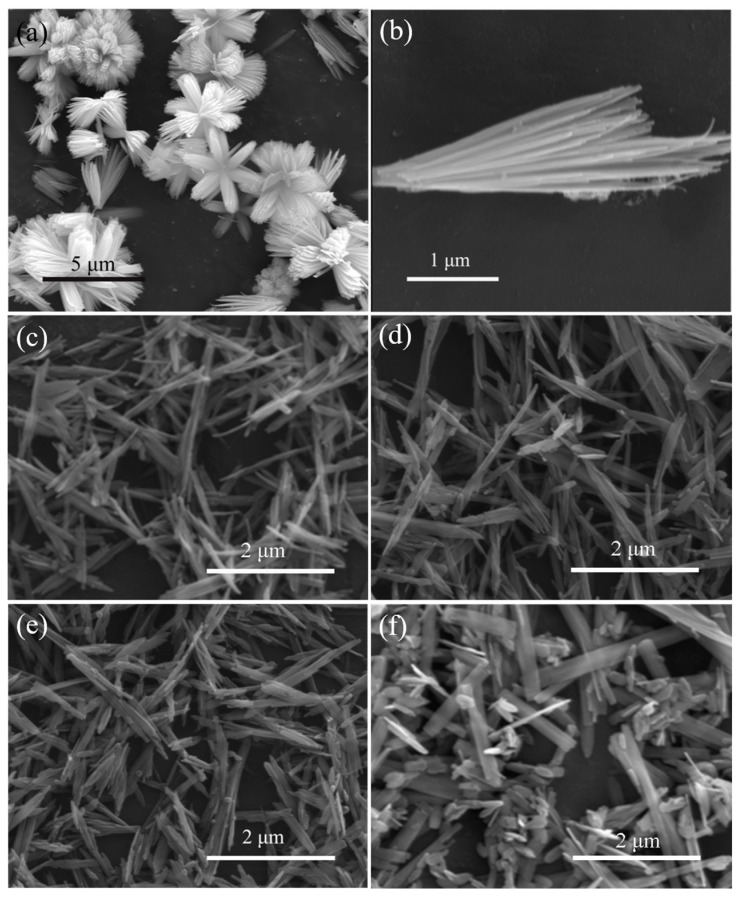
SEM images of (**a**,**b**) pure ZnOHF, (**c**) 0.1 at.% Al-ZnOHF, (**d**) 0.2 at.% Al-ZnOHF, (**e**) 0.5 at.% Al-ZnOHF, (**f**) 1 at.% Al-ZnOHF.

**Figure 3 materials-16-03577-f003:**
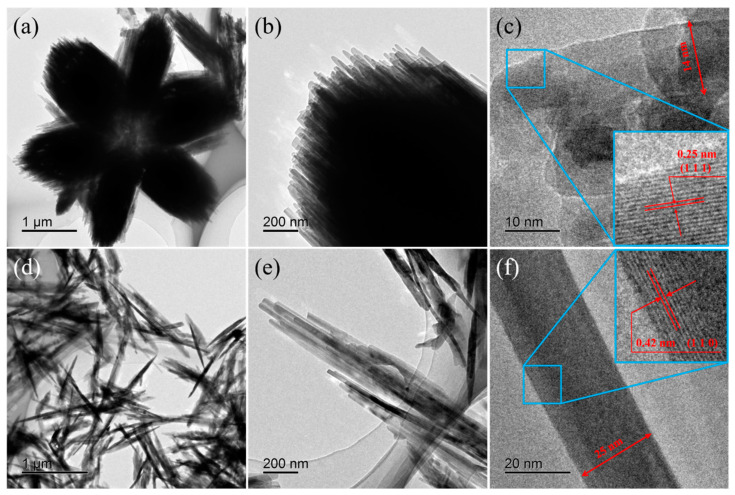
TEM and HRTEM images of (**a**–**c**) pure ZnOHF, (**d**–**f**) 0.5 at.% Al-ZnOHF.

**Figure 4 materials-16-03577-f004:**
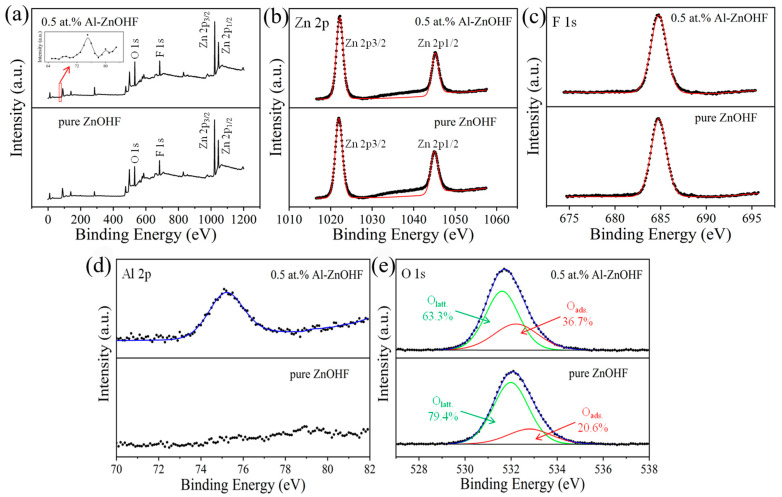
(**a**) XPS wide spectrum, (**b**) Zn 2p region, (**c**) F 1s region, (**d**) Al 2p region, and (**e**) O 1s region XPS spectra of pure ZnOHF and 0.5 at.% Al-ZnOHF. (the dots are test data and the red and blue lines are fixed curves).

**Figure 5 materials-16-03577-f005:**
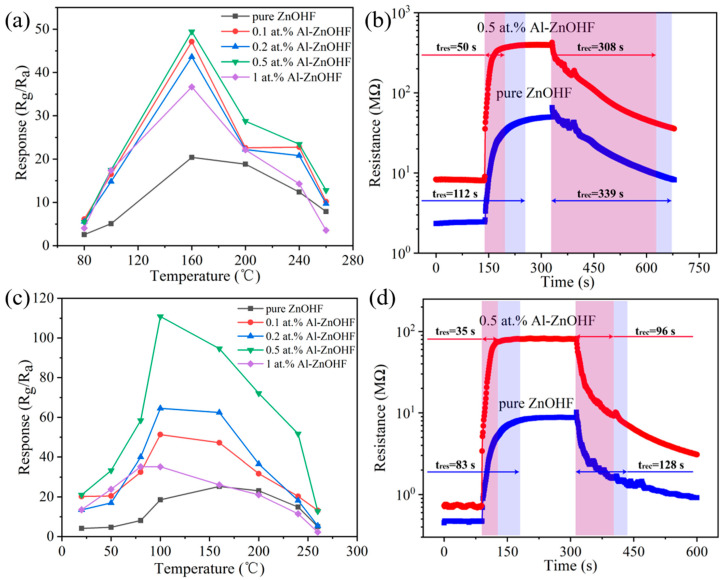
(**a**) Gas responses of all obtained samples to 10 ppm NO_2_ at different temperatures in the dark. (**b**) The response–recovery curves of pure ZnOHF and 0.5 at.% Al-ZnOHF to 10 ppm NO_2_ at 160 °C in dark. (**c**) Gas responses of all obtained samples to 10 ppm NO_2_ at different temperatures under UV assistance. (**d**) The response–recovery curves of pure ZnOHF and 0.5 at.% Al-ZnOHF to 10 ppm NO_2_ at 100 °C under UV assistance.

**Figure 6 materials-16-03577-f006:**
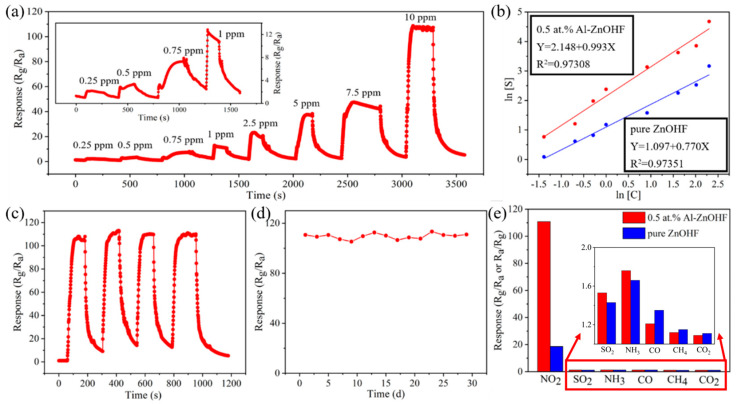
Under UV assistance and 100 °C: (**a**) the dynamic response curve of 0.5 at.% Al-ZnOHF; (**b**) the linear relationship between the logarithm of response value and the logarithm of gas concentration of pure ZnOHF (blue) and Al-ZnOHF (red); (**c**) response–recovery curve in 4 cycles of 0.5 at.% ZnOHF; (**d**) the response change of 0.5 at.% ZnOHF at 1 month, and (**e**) response value of pure ZnOHF and 0.5 at.% Al-ZnOHF to 10 ppm NO_2_, SO_2_, NH_3_, CO, CH_4_ and CO_2_.

**Figure 7 materials-16-03577-f007:**
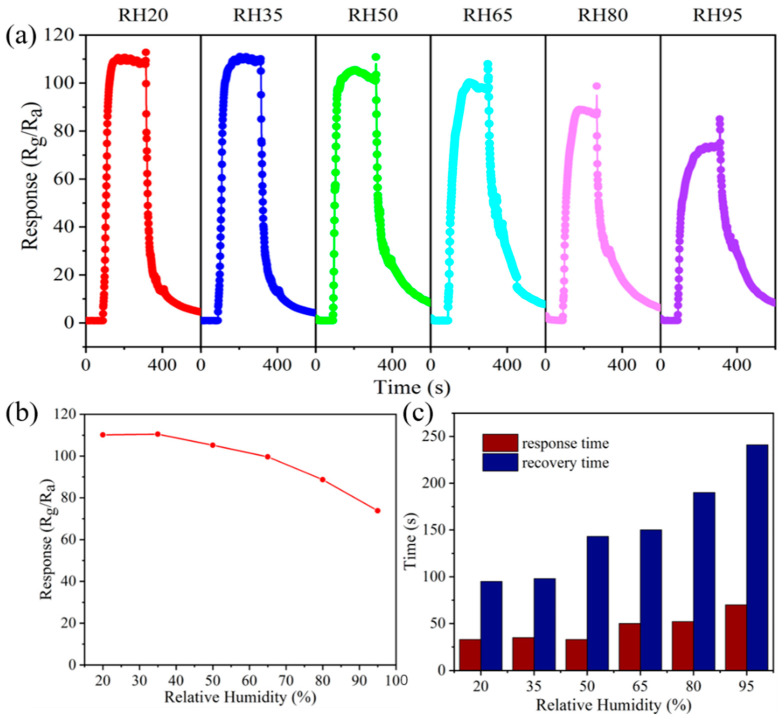
(**a**) The response–recovery curves, (**b**) the response value change, (**c**) response/recovery time of 0.5 at.% Al-ZnOHF to 10 ppm NO_2_ at various relative humidities at 100 °C under UV radiation.

**Figure 8 materials-16-03577-f008:**
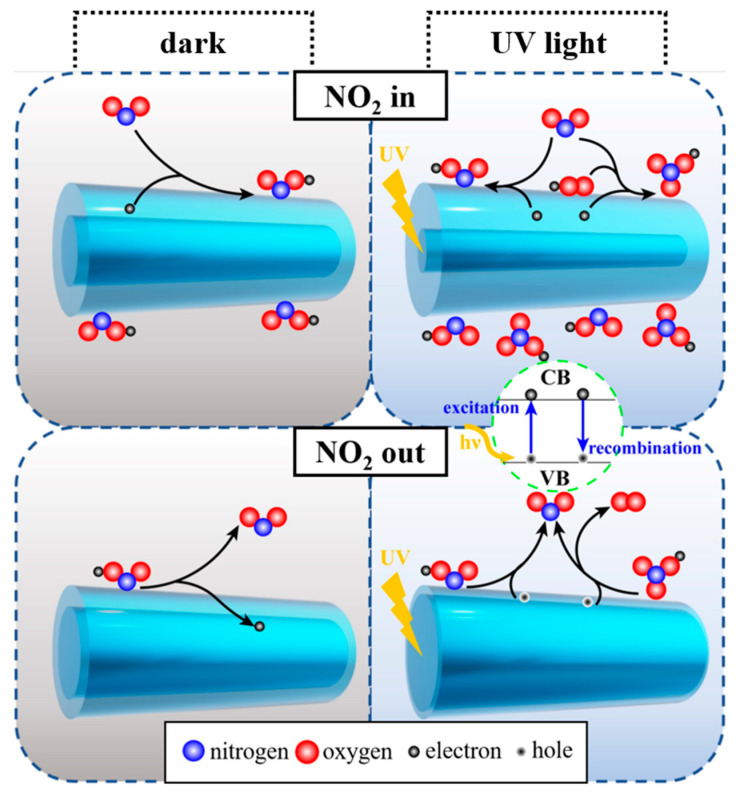
The response and recovery process of Al-ZnOHF in dark and with UV assistance.

**Figure 9 materials-16-03577-f009:**
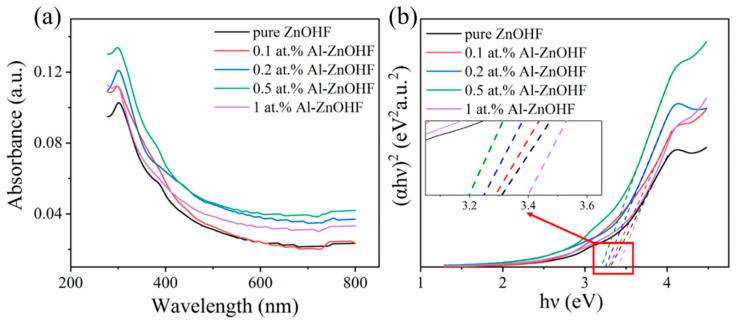
(**a**) The UV-vis diffuse reflectance spectra and (**b**) T-plots of (αhν)^2^ vs. hν of all as-prepared samples (the dashed line is the tangent line of curves).

**Table 1 materials-16-03577-t001:** The NO_2_-sensing performance of materials in references and this work.

Materials	Operating Condition	S (R_g_/R_a_ or R_a_/R_g_)	T_res_/T_rec_ (s)	Detection Limit	Ref.
ZnO/SnO_2_ composite	40 °C, UV (395 nm, 34 μW/cm^2^)	25 (1 ppm)	251/470	100 ppb	[40]
Ag-ZnO nanoparticles	25 °C, visible light (455 nm, 70 mW/cm^2^)	2.5 (5 ppm)	200/175	1 ppm	[41]
Si-ZnO thin films	75 °C, purple-blue (430 nm)	19.1 (400 ppb)	60/180	-	[42]
CeO_2_/ZnO nanorods	120 °C	190.6% (5 ppm)	104/417	100 ppb	[43]
Fe_2_O_3_-ZnO nanostructures	300 °C	6.34 (10 ppm)	26/185	1 ppm	[44]
3D flower-like ZnOHF	200 °C	82.71 (10 ppm)	13/35	0.1 ppm	[14]
0.5 at.%Al-ZnOHF	100 °C, UV (395 nm, 6 mW/cm^2^)	110.83 (10 ppm)	35/96	0.25 ppm	This work

## Data Availability

The data presented in this study are available on request from the corresponding author. The data are not publicly available due to privacy restrictions.

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
