# Peer review of "Highly Efficient NO2 Sensors Based on Al-ZnOHF under UV Assistance"

_materials, 2023, doi:10.3390/ma16093577_

Round 1
Reviewer 1 Report
This work demonstrates a highly efficient NO2 gas sensor based on Al-doped ZnOHF under UV-assistance. Some concerns must be answered before being accepted for publication in this journal.
1. What is the novelty of this work when compared to Reference 13?
2. Some groups provided an alternative way of improving gamma-ray response which can be compared in the introduction along with reference 16 (UV-radiation).
a. https://doi.org/10.1016/j.snb.2022.132255
b. https://doi.org/10.1016/j.jhazmat.2021.125841
3. On Page 2, line 80, what does the term "strength" signifies here?
4. On Page 3, line 98, Keep all the units in SI format.
5. On page 3, line 114, does the term "salary" correct?
6. What is the reason behind waiting for 7 days before the gas sensing test?
7. Why does the intensity of the XRD peak gets enhanced after doping?
8. In figure 1b, is there any possibility of two features around the same region? Because at higher doping concentrations the peak becomes very broad.
9. In figure 2, for comparison purposes, the resolution of the all the SEM images at different doping concentrations must be the same.
10. Why does the author compare the d-spacing of the (111) peak in pure films and the (110) peak in the case of 0.5 %Al doped?
11. In Figure 7e, show the enlarged values of other gases in the inset of the same figure.
12. Keep the number of the equations in order correct throughout the manuscript.
13. The authors must add their previous work results in Table 1 for comparison.
This work demonstrates a highly efficient NO2 gas sensor based on Al-doped ZnOHF under UV-assistance. Some concerns must be answered before being accepted for publication in this journal.
1. What is the novelty of this work when compared to Reference 13?
2. Some groups provided an alternative way of improving gamma-ray response which can be compared in the introduction along with reference 16 (UV-radiation).
a. https://doi.org/10.1016/j.snb.2022.132255
b. https://doi.org/10.1016/j.jhazmat.2021.125841
3. On Page 2, line 80, what does the term "strength" signifies here?
4. On Page 3, line 98, Keep all the units in SI format.
5. On page 3, line 114, does the term "salary" correct?
6. What is the reason behind waiting for 7 days before the gas sensing test?
7. Why does the intensity of the XRD peak gets enhanced after doping?
8. In figure 1b, is there any possibility of two features around the same region? Because at higher doping concentrations the peak becomes very broad.
9. In figure 2, for comparison purposes, the resolution of the all the SEM images at different doping concentrations must be the same.
10. Why does the author compare the d-spacing of the (111) peak in pure films and the (110) peak in the case of 0.5 %Al doped?
11. In Figure 7e, show the enlarged values of other gases in the inset of the same figure.
12. Keep the number of the equations in order correct throughout the manuscript.
13. The authors must add their previous work results in Table 1 for comparison.
Reviewer 2 Report
The article "Highly-efficient NO2 Sensors Based on Al-ZnOHF under UV 2 Assistance" belongs to the field of chemical gas sensing which is an important applied area of science. In general, the work is well written, contains modern methods of physico-chemical analysis and gas-sensor measurements are carried out in full. The presented sensing material ZnOHF is a new and interesting object. Nevertheless, I have a number of questions which need to be clarified, before publication in "Materials":
1) The XPS spectra are important, therefore I suggest to move fig.S1 to the main part.
2) I suggest that in addition to fig.S2, data on photoresponse (change in R under UV exposure in abs. units) on temperature be given. These data would be useful, they can also correlate well with those already available.
3) ZnOHF is not the most trivial material for chemoresistive gas sensors. Its thermal stability is not very clear from the text of the article, given that measurements were made at high temperatures (up to 250 C). In this regard I suggest that further investigations should be carried out. It is necessary to investigate the material by means of thermal analysis (DSC-TGA) and to show its thermal behaviour. Also, I would recommend to give data on its phase and chemical composition after the sensor measurements.
4) The following articles can be used to extend the literature review of this work: 10.3390/bios13040445, 10.1016/j.apsusc.2022.152974.
Round 2
Reviewer 1 Report
Accept
Reviewer 2 Report
The authors' answers are convincing, the article can be published.